# Serum Proteome Alterations in Human Cystathionine β-Synthase Deficiency and Ischemic Stroke Subtypes

**DOI:** 10.3390/ijms20123096

**Published:** 2019-06-25

**Authors:** Marta Sikora, Izabela Lewandowska, Małgorzata Kupc, Jolanta Kubalska, Ałła Graban, Łukasz Marczak, Radosław Kaźmierski, Hieronim Jakubowski

**Affiliations:** 1European Centre for Bioinformatics and Genomics, Institute of Bioorganic Chemistry, Polish Academy of Sciences, 60-965 Poznań, Poland; martas@ibch.poznan.pl (M.S.); izek1988@gmail.com (I.L.); lukasmar@ibch.poznan.pl (Ł.M.); 2Department of Biochemistry and Biotechnology, University of Life Sciences, 60-632 Poznań, Poland; gosia.kupc@gmail.com; 3Department of Genetics, Institute of Psychiatry and Neurology, 02-957 Warsaw, Poland; kubalska@ipin.edu.pl; 4First Department of Neurology, Institute of Psychiatry and Neurology, 02-957 Warsaw, Poland; graban@ipin.edu.pl; 5Department of Neurology and Cerebrovascular Disorders, Poznań University of Medical Sciences, L. Bierkowski Hospital, 60-631 Poznań, Poland; rkazmierski@ump.edu.pl; 6Department of Microbiology, Biochemistry and Molecular Genetics, Rutgers-New Jersey Medical School, International Center for Public Health, Newark, NJ 07-103, USA

**Keywords:** cystathionine β-synthase deficiency, ischemic stroke subtypes, proteostasis, proteomics, label-free mass spectrometry, serum protein markers

## Abstract

Ischemic stroke induces brain injury via thrombotic or embolic mechanisms involving large or small vessels. Cystathionine β-synthase deficiency (CBS), an inborn error of metabolism, is associated with vascular thromboembolism, the major cause of morbidity and mortality in affected patients. Because thromboembolism involves the brain vasculature in these patients, we hypothesize that CBS deficiency and ischemic stroke have similar molecular phenotypes. We used label-free mass spectrometry for quantification of changes in serum proteomes in CBS-deficient patients (*n* = 10) and gender/age-matched unaffected controls (*n* = 14), as well as in patients with cardioembolic (*n* = 17), large-vessel (*n* = 26), or lacunar (*n* = 25) ischemic stroke subtype. In CBS-deficient patients, 40 differentially expressed serum proteins were identified, of which 18 were associated with elevated homocysteine (Hcy) and 22 were Hcy-independent. We also identified Hcy-independent differentially expressed serum proteins in ischemic stroke patients, some of which were unique to a specific subtype: 10 of 32 for cardioembolic vs. large-vessel, six of 33 for cardioembolic vs. lacunar, and six of 23 for large-vessel vs. lacunar. There were significant overlaps between proteins affected by CBS deficiency and ischemic stroke, particularly the cardioembolic subtype, similar to protein overlaps between ischemic stroke subtypes. Top molecular pathways affected by CBS deficiency and ischemic stroke subtypes included acute phase response signaling and coagulation system. Similar molecular networks centering on NFκB were affected by CBS deficiency and stroke subtypes. These findings suggest common mechanisms involved in the pathologies of CBS deficiency and ischemic stroke subtypes.

## 1. Introduction

Stroke is one of the leading causes of morbidity and mortality in the world [1]. Overall stroke burden increased across the globe in both men and women of all ages [2]. Unlike cardiovascular disease, which ultimately involves vascular thrombosis, ischemic stroke induces a focal injury in the brain with thrombotic or embolic mechanisms (from cardiac source or periphery). Proper treatment relies on the differentiation between ischemic and hemorrhagic stroke [3,4]. There are three major subtypes of ischemic stroke: cardioembolic, large-vessel, and lacunar [1]. Although the heterogeneity of stroke was a subject of numerous studies, the therapeutic interventions are limited [3,4].

The individuals with cystathionine β-synthase (CBS) deficiency, a rare inborn error of metabolism, have 50% chance of vascular thromboembolism by the age of 30, which is the major cause of morbidity and mortality [5]. Although little is known regarding the mechanisms leading to the onset of ischemic stroke in CBS-deficient individuals [6], 32% of those patients suffer thromboembolic stroke [5]. Other vascular thromboembolic incidents in CBS-deficient patients occur in the heart (4%), peripheral veins (51%), and arteries (11%). Whether and how strokes in CBS-deficient patients are related to strokes in the general population is not known.

An important goal in stroke research is the dissection of mechanisms involved in subtypes of ischemic stroke. One approach to elucidating molecular mechanisms underlying the ischemic stroke is to identify proteins whose expression is altered by the ischemic stroke subtype. It is likely that different subtypes of ischemic stroke cause specific alterations in the serum proteome that contribute to the cerebrovascular pathology. Defining proteins affected by each ischemic stroke subtype will help identify molecular pathways involved in the stroke pathology and uncover potential pharmacologic targets for preventing or treating the disease. An analysis of proteins and biological pathways affected by CBS deficiency and ischemic stroke subtypes was not previously reported. For these reasons, the present work was undertaken to ascertain proteomic signatures associated with CBS deficiency and ischemic stroke subtypes. We accomplished this aim by using label-free mass spectrometry and bioinformatic analyses to study changes in serum proteomes associated with CBS deficiency and ischemic stroke subtypes and to identify molecular pathways and networks involved.

## 2. Results

Characteristics of CBS-deficient patients and unaffected control individuals are shown in Table 1. Corresponding data for patients with different ischemic stroke subtypes are shown in Table 2. 

Label-free nanoLC–MS/MS identified 196–198 serum proteins with a minimum of two peptides and 1% false discovery rate (FDR) in each group of subjects. The Proteome Discoverer (PD) analysis showed that the overlap between the duplicate injections was >90% at the protein level. Principal component analysis (PCA) differentiated between CBS-deficient patients and control individuals (Figure 1A). In ischemic stroke patients, PCA differentiated between cardioembolic and lacunar stroke (Figure 1B), but showed similarities between cardioembolic and large-vessel stroke subtypes, as well as between large-vessel and lacunar stroke subtypes (Figure 1B).

### 2.1. Serum Proteins Affected by CBS Deficiency

In CBS-deficient patients, we identified 40 serum proteins whose levels were significantly different from unaffected controls (Figure 2A). Fifteen proteins were significantly upregulated (1.11- to 3.39-fold) and 25 were significantly downregulated (0.57- to 0.93-fold) (Table 3). Most of the proteins affected by CBS deficiency include those involved in acute phase response (SAA1, SERPINA1, ORM2, AHSG), immune response (IGHD, IGJ, APC, C4BPA, IGHV3-72, IGK, IGKV2D-24), blood coagulation (FBLN1, KNG, APOH, C1S, C1R, CF1, CBP2, SERPINC1, SERPINF2, SERPIND1, F2, F13B), and lipid/cholesterol transport/metabolism (APOC1, APOC3, APOA1, APOM) (Table 3). Of the 40 CBS-responsive proteins, 18 were specific for CBS deficiency (Table 4), while 22 were also affected by ischemic stroke subtypes (Table 3). 

### 2.2. Serum Proteins Affected by Ischemic Stroke Subtype

We identified serum proteins whose levels were significantly affected by ischemic stroke subtypes (Table 3). There were 32 proteins with different levels between cardioembolic and large-vessel stroke. Of those, levels of 11 proteins were elevated (from 1.12-fold for C1QB to 6.31-fold for SAA1) and 21 were reduced (from 0.92-fold for KNG1 to 0.44 for C4BPA) in cardioembolic vs. large-vessel stroke. 

Levels of 33 proteins were different between cardioembolic and lacunar stroke: 13 proteins were elevated (from 1.12-fold for ITIH4 to 11.13-fold for SAA1) and 20 proteins were reduced (from 0.92-fold for KNG1 to 0.35-fold for histone H4) in cardioembolic vs. lacunar stroke (Table 3).

There were also 23 proteins that differentiated large-vessel and lacunar stroke: 12 proteins were elevated (from 1.09-fold for PLG to 2.41-fold for CRP) and 11 proteins were reduced (from 0.91-fold for N-PGLYRP2 to 0.38-fold for PZP) (Table 3). 

The majority of proteins affected in the ischemic stroke subtype comparisons are involved in acute phase response (SAA1, CRP, SERPINA3, SERPINA1, LBP, ORM1, AHSG), immune/inflammatory response (APC, C4BPA, IGK, IGKV1V-12, PGLYRP2), blood coagulation (A2M, FBLN1, FGA, PLG, SERPIND1, F13B, etc.), and cholesterol transport/metabolism (APOC1, APOC3, APOL1, etc.) (Table 3).

### 2.3. Ischemic Stroke Subtype-Specific Proteins

Some of the serum proteins affected by ischemic stroke were unique to a specific subtype: 10 of 32 for cardioembolic vs. large-vessel, six of 33 for cardioembolic vs. lacunar, and six of 23 for large-vessel vs. lacunar (Figure 2B, Table 4). 

Some of the stroke subtype-specific proteins (FBLN1, F2, SERPINF2, CBP2, FCN3, GPX3, IKG, and GSN) were also affected by the CBS deficiency (Table 3).

Other proteins were shared only between the stroke subtype comparisons. For example, 19 differentiating proteins were shared among cardioembolic vs. large-vessel stroke and cardioembolic vs. lacunar stroke comparisons; 17 differentiating proteins were shared between cardioembolic vs. lacunar stroke and large-vessel vs. lacunar stroke comparisons; nine differentiating proteins were shared between large-vessel vs. lacunar and cardioembolic vs. large-vessel stroke comparisons. There were six common proteins shared between all three stroke subtype comparisons (Figure 2B, Table 3). Remarkably, four of those proteins, SERPINA3 and CRP involved in acute-phase response, and FGA and PLG involved in blood coagulation, were not affected by CBS deficiency.

### 2.4. Overlap between Proteins Affected by CBS Deficiency and Ischemic Stroke Subtype

Most of the 40 proteins affected by the CBS deficiency were also affected in stroke subtype comparisons: 15 for cardioembolic vs. large-vessel stroke, 13 for cardioembolic vs. lacunar stroke, and seven for large-vessel vs. lacunar stroke. Of the 40 proteins, 18 were CBS deficiency-specific, not affected by the stroke subtype (Figure 2A).

### 2.5. Validation of Label-Free Mass Spectrometry Analyses by ELISA

To validate label-free mass spectrometry experiments, we quantified three of the differentiating proteins by ELISA. The pro-inflammatory protein SAA1 that was upregulated in cardioembolic vs. large-vessel stroke and cardioembolic vs. lacunar stroke in the label-free experiments (Table 3) was also upregulated in ELISA experiments (2.41-fold and 4.70-fold, respectively). The acute phase response protein SERPINC1, downregulated in CBS-deficient patients, and the oxidative stress protein GPX3, upregulated in CBS-deficient patients and large-vessel vs. lacunar stroke in the label-free experiments (Table 3), were similarly affected in ELISA experiments (0.63-fold and 1.19-fold, respectively).

### 2.6. Bioinformatic Analyses

To identify biological pathways linked to proteins affected by CBS deficiency or stroke subtype, we carried out bioinformatic analyses using IPA resources. We found that proteins affected by CBS deficiency were significantly enriched in 13 molecular pathways, while proteins affected by stroke subtypes were similarly enriched in the same molecular pathways (Figure 2C). The top four highly significant (−log(*p*-value) = 20–31) pathway categories affected by both CBS deficiency and stroke subtypes were LXR/RXR activation, acute phase response signaling, FXR/RXR activation, and coagulation system (Figure 2C). 

IPA identified similar sets of biological networks for CBS deficiency and ischemic stroke subtypes. The two top-scoring networks (score = 35–60) associated with CBS deficiency were “humoral immune response, inflammatory response, developmental disorder” (Figure 3A) and “metabolic disease, hematological system development and function, lipid metabolism” (Table 5). A predominant network, with a score of 60, associated with CBS deficiency, contained 34 proteins: 24 were quantified by label-free mass spectrometry and 10 identified by IPA to interact with the quantified proteins (Figure 2A). Proteins in these networks show strong interactions with NFκB.

The two top-scoring networks for cardioembolic vs. large-vessel stroke (score = 33–36) were “hematological system development and function, cell-to-cell signaling and interaction, organismal functions” (Figure 3B) and “metabolic disease, developmental disorder, hereditary disorder” (Table 5). The two top-scoring networks for cardioembolic vs. lacunar stroke (score = 32–50) were “cell-to-cell signaling and interaction, hematological system development and function, neurological disease” (Figure 3C) and “inflammatory response, metabolic disease, cell-to-cell signaling and interaction” (Table 5). The two top-scoring networks for large-vessel vs. lacunar stroke (score = 24 each) were “inflammatory response, cellular movement, immune cell trafficking” (Figure 3D) and “cell-to-cell signaling and interaction, hematological system development and function, inflammatory response” (Table 5). 

A predominant network, with a score of 36, associated with cardioembolic vs. large-vessel stroke, contained 35 proteins: 16 were quantified by label-free mass spectrometry and 19 identified by IPA to interact with the quantified proteins (Figure 3B, Table 5). A similar predominant network with a score of 50, containing 40 proteins (21 quantified by label-free mass spectrometry and 19 identified by IPA), was associated with cardioembolic vs. lacunar stroke (Figure 3C, Table 5). Another similar network with a score of 24, containing 34 proteins (11 quantified by label-free mass spectrometry and 23 identified by IPA) was associated with large-vessel vs. lacunar stroke (Figure 3D, Table 5). Proteins in these networks show strong interactions with NFκB.

### 2.7. Involvement of Homocysteine and Anti-N-Hcy-Protein Autoantibodies

CBS deficiency is characterized by the accumulation of homocysteine (Hcy) [5] and its metabolites, Hcy-thiolactone [7] and *N*-Hcy-protein [8], each of which induces metabolite-specific pro-atherogenic changes in gene expression in human endothelial cells [9,10]. Our previous work showed that *N*-Hcy-protein is autoimmunogenic and that anti-*N*-Hcy-protein autoantibodies were elevated in serum of stroke patients [10,11]. To determine whether changes in the serum proteomes (Table 3) can be caused by HHcy, we quantified serum total Hcy (tHcy) and anti-*N*-Hcy-protein autoantibody levels. We found that levels of anti-*N*-Hcy-protein autoantibodies (0.33 ± 0.16 vs. 0.12 ± 0.10 A_492_, *p* = 0.005) and tHcy (71.2 ± 55.6 vs. 9.7 ± 5.9 μM, *p* < 0.001) were significantly elevated in CBS-deficient patients relative to unaffected controls (Table 1). The 2.75-fold increase in anti-*N*-Hcy-protein autoantibodies reflects the 3.9-fold elevation in *N*-Hcy-Lys525-albumin [12] and 6.2-fold elevation in total *N*-Hcy-protein [8] in CBS-deficient patients. 

In contrast, in ischemic stroke patients, we found that anti-*N*-Hcy-protein autoantibody levels were not signifcantly different betwen subtypes: 0.12 ± 0.08, 0.14 ± 0.11, and 0.10 ± 0.07 A_492_ in cardioembolic, lacunar, and large-vessel stroke, respectively (Table 2). Corresponding tHcy levels were 3.5 ± 1.6, 3.0 ± 2.0, and 3.3 ± 1.2 μM, respectively, which were also not significantly different (Table 2). Earlier studies, with one exception [13], also reported that tHcy levels did not differ between stroke subtypes [14,15,16,17]. 

## 3. Discussion

We used label-free mass spectrometry to identify serum proteins whose levels were affected by CBS deficiency and ischemic stroke subtypes. We found that many of the proteins affected by CBS deficiency were also affected in ischemic stroke patients. Our results suggest that CBS deficiency and ischemic stroke, particuraly the cardioembolic stroke subtype, share similar molecular mechanisms. Both pathologies affect molecular networks that contain proteins showing strong interactions with NFκB.

Our findings in CBS-deficient patients can also explain the cardiovascular and neurological pathologies, including the pro-thrombotic phenotype, often affecting the brain vasculature in these individuals. Specifically, the identification of CBS deficiency-responsive proteins that are involved in acute phase response (e.g., SAA1, SERPINA1, ORM2, AHSG), immune response (e.g., IGHD, IGJ, APC, C4BPA, IGHV3-72, IGK, IGKV2D-24), blood coagulation (e.g., FBLN1, KNG, APOH, C1S, C1R, CF1, CBP2, SERPINC1, SERPINF2, SERPIND1, F2, F13B), and lipid/cholesterol transport/metabolism (APOC1, APOC3, APOA1, APOM) supports this conclusion. 

Present findings in ischemic stroke patients highlight the differences in proteomic signatures between stroke subtypes. These include ten proteins (APCS, APOM, C1qA, C4BPA, CBP2, F2, FBLN1, IGKV1D-12, KLKB1, SERPINF2) that differentiate the cardioembolic vs. large-vessel stroke, six other proteins (AMBP, APOA4, FCN3, ITIH4, LBP, PF4) that differentiate the cardioembolic vs. lacunar stroke, and another six proteins (APOL1, C5, GPX3, GSN, H2AFJ, IGK) that differentiate the large-vessel vs. lacunar stroke (Table 3, Figure 3B). To the best of our knowledge, with the exception of C5, known to be associated with large-vessel stroke in a Chinese population [18], these differentiating proteins, weree not previously associated with a specific subtype of the ischemic stroke. These findings raise a possiblity that the ischemic stroke subtype-specific proteins can be exploited, most likely as a panel, in diagnosis and prognosis of outcomes, and possibly therapeutic interventions. Although previous studies identified several other protein biomarkers of ischemic stroke, none had sufficient evidence to support use in clinical settings [19]. For these reasons, protein biomarkers of ischemic stroke subtypes identified in the present study warrant validation in larger cohorts.

Comparisons of proteomes between CBS deficiency, on the one hand, and ischemic stroke subtypes, on the other, indicate that sevaral proteins (Figure 2A) and molecular pathways (Figure 2C) are shared between these pathologies (Table 3 and Table 4). However, of the CBS-associated proteins (*n* = 40) about half (*n* = 18) were affected only by CBS deficiency (Table 3, Figure 2A). These CBS deficiency-specific proteins might be associated with HHcy-related pathologies affecting other organ systems, e.g., the heart, lungs, bones, or skin, in addition to the brain [5]. Bioinformatic analysis shows that CBS deficiency affects molecular pathways and networks, centering on NFκB, which affect blood clotting, immune response, and inflammation (Table 5, Figure 3A). As shown in the present work (the last paragraph of Section 2), the immune response in CBS-deficient patients involves also autoantibodies against *N*-Hcy-protein [11]. Our previous work showed that *N*-Hcy-proteins, including prothrombotic *N*-Hcy-fibrinogen, accumulate in CBS-deficient patients [8]. In addition to elevated total *N*-Hcy-protein, CBS-deficient patients also have elevated pro-thrombotic *N*-Hcy-fibrinogen. Thus, the dysregulated proteostasis involving pro-thrombotic *N*-Hcy-fibrinogen and other proteins participating in blood coagulation, identified in the present study, most likely contributes to the pro-thrombotic phenotype that could explain why four of the 10 CBS-deficient patients involved in our study had a stroke at a young age.

Notably, we also found that more proteins were shared between the cardioembolic stroke subtype and CBS deficiency (13–15 shared proteins) than between the large-vessel or the lacunar subtype and CBS deficiency (seven shared proteins) (Figure 2A), suggesting that the CBS deficiency proteome better resembles the cardioembolic stroke proteome than the large-vessel or the lacunar stroke proteome. This conclusion is consistent with findings of other investigators showing that CBS-deficient patients are prone to the embolic stroke [5,6].

Only four out of 10 CBS-deficient patients suffered ischemic strokes, which ocurred long before their participation in the present study. Yet, the CBS-deficient patients exhibited a pro-stroke proteomic signature that was very similar to the proteomic signature of the ischemic stroke patients, each of whom suffered an ischemic stroke immediately before participation in the study. Thus, CBS deficiency causes chronic dysregulation of the proteostasis in the circulation, which affects the blood-clotting system and makes the CBS-deficient patients prone to ischemic strokes.

Our study also highlights subtle molecular differences between CBS deficiency and ischemic stroke subtupes. For example, some of the differentiation proteins, such as SERPINA3 and CRP involved in inflammatory/acute-phase response, as well as FGA and PLG involved in blood coagulation, were associated with the cardioembolic, large-vessel, and lacunar ischemic stroke subtypes, but not with CBS deficiency (Table 3). Bioinformatic analyses showed that ischemic stroke subtypes were associated with similar molecular processes and networks, centering on NFκB and affecting inflammation and blood clotting (Table 5, Figure 3B–D). 

Some of the changes in the CBS deficiency proteome (i.e., those involving the 18 CBS deficiency-specific proteins; Figure 2A, Table 3) were most likely caused by HHcy because Hcy and anti-*N*-Hcy-protein autoantibody levels were elevated in CBS-deficient patients. Thus, our findings suggest two effects of CBS deficiency: HHcy-related, which affected 18 proteins, and HHcy-independent, which affected 22 other proteins. The 22 HHcy-independent proteins were also affected in patients with different ischemic stroke subtypes, in whom Hcy and anti-*N*-Hcy-protein autoantibody levels were similar.

Our present findings that CRP and FGA levels were associated with the three ischemic stroke subtypes confirm previous findings in other cohorts of stroke patients. For example, CRP was elevated in patients with atherothrombotic or cardioembolic stroke. Patients with atherothrombotic stroke had significantly higher CRP relative to the lacunar or idiopathic strokes [20]. Fibrinogen was elevated in cardioembolic stroke in one study [20] and in the large-vessel vs. small vessel stroke in another study [21].

Several other proteins identified in the present study as differentiating between ischemic stroke subtypes were previously implicated in stroke or vascular outcomes. For example, SERPINA1, whose levels we found to be increased in CBS deficiency, as well as in the large-vessel vs. lacunar stroke (Table 3), has a coding variant that increases the risk for large vessel stroke [22,23]. In another study, low levels of SERPINA1, PLG, KLKB1, ITIH4, and APOL1 in serum of patients with lacunar stroke were associated with poor prognosis and increased risk of recurrent stroke or myocardial infarction and cognitive decline [24].

Reduced levels of SERPINC1 that we found in the CBS-deficient patients in the present study (Table 3) were also observed in Han Chinese ischemic stroke patients [25]. Similarly, KLKB1, which was reduced in the cardioembolic vs. large-vessel stroke and in the CBS-deficient patients, was found to be reduced in Han Chinese ischemic stroke patients [25]. However, ITIH4, which was elevated in cardioembolic vs. lacunar stroke in the present study (Table 3), was found to be reduced in Han Chinese ischemic stroke patients [25]. Elevated CRP and SAA1, which we observed in the ischemic stroke subtype comparisons or in CBS deficiency (Table 3), were also observed in Han Chinese ischemic stroke patients [25] and in cardioembolic and large-vessel childhood arterial ischemic stroke [26]. GSN, which we found to be reduced in the large-vessel vs. lacunar stroke comparison (Table 3), was also reduced in Chinese ischemic stroke patients [27]. Reduced AFM and TTR, which we found in the cardioembolic vs. large-vessel stroke comparison and in the CBS deficiency (Table 3), were also observed in Han Chinese ischemic stroke patients [25]. Notably, reduced TTR is associated with poor prognosis in ischemic stroke patients [28].

Levels of serum APOC1, which were similarly affected by CBS deficiency and ischemic stroke subtypes, and levels APOC3, which were affected only by CBS deficiency in the present study (Table 3), were also reported to be similar between ischemic stroke and controls but significantly different between hemorrhagic stroke and controls in another study [29]. Taken together, these findings suggest that APOC1 and APOC3 can be markers of hemorhagic stroke, but not of the ischemic stroke subtypes. Interestingly, our findings indicate that APOA1 is also not related to ischemic stroke subtypes. However, a combination of APOC3 and APOA1 markers improved the differentiation between hemorrhagic and ischemic stroke [4]. 

Some limitations of the present study have to be considered. Firstly, the differentiating proteins identified in the present study may promote stroke or be a feature present in patients with CBS deficiency and in patients with ischemic stroke. However, because many of the dysregulated proteins identified in the present study are involved in blood coagulation, changes in these proteins are likely to be prothrombotic and promote stroke. Secondky, CBS-deficient patients (37.3 ± 7.3 years) were much younger than ischemic stroke patients (67.5 ± 12.4 years). This, however, was unavoidable because CBS-deficient patients have strokes at an early age and suffer premature mortality. Furthermore, our findings that over half (22 of 40, Figure 2A) of differentiating proteins were shared between CBS-deficient and ischemic stroke patients indicate that these proteins are associated with stroke rather than with age. Thirdly, to identify ischemic stroke subtype-related proteins, we compared protein levels between cardioembolic, large-vessel, and lacular stroke subtypes without a healthy control group as a reference. An alternative approach is to compare protein levels between each stroke subtype group and a healthy control group as a reference. However, both approaches yield valid results and were used in previous studies. For example, other investigators analyzed differences in plasma protein levels between stroke subtypes by using another stroke subtype as a reference, and they successfully identified proteins (FBG and inflammatory markers) differentiating between stroke subtypes [21,26]. Furthermore, the fact that the present study identified many proteins that were previously known to be associated with ischemic stroke indicates that our data are reliable. In addition, the validity of our study is supported by the findings that many differentiating proteins were shared between the stroke subtypes part and the CBS-deficiency part of the study, which did have a healthy, age- and sex-matched group as a reference. Fourthly, the number of subjects in each studied group was relatively small due to the stringent selection criteria used to eliminate possible confounding by other diseases. 

In conclusion, our study revealed previously unanticipated similarties between proteomic signatures of CBS deficiency and ischemic stroke subtypes, suggesting that similar mechanisms are involved in both pathologies. We also identified proteins that were not previously associated with a specific subtype of the ischemic stroke. A validation in larger cohorts is required in order to assess the generality of the present findings.

## 4. Materials and Methods 

### 4.1. CBS-Deficient Patients

The study included serum samples from previously described Danish [30] (*n* = 4) and Polish [31,32] (*n* = 6) CBS-deficient patients (mean age 37.3 ± 7.3 years; five males, five females) and gender- and age-matched healthy unaffected individuals (*n* = 14). Two of the Polish CBS-deficient patients had a stroke before diagnosis in childhood at age 3–12. Danish CBS-deficient patients were diagnosed at ages 7, 21, 22, and 36; two of them had episodic transient ischemic attacks at ages 22 and 36. All patients were on a homocysteine (Hcy)-lowering treatment (vitamin B_6_). Characteristics of the CBS-deficient patients are described in Table 1.

### 4.2. Ischemic Stroke Patients

Ischemic stroke patients (*n* = 68, 67.5 ± 12.4 years old; 46% female) included in the study were selected from consecutive 161 stroke patients treated at the Department of Neurology, University of Medical Sciences Hospital, Poznań. Individuals with hemorrhagic stroke, or with uncertain or other causes were excluded. Patients with ischemic stroke who had other diseases including a malignant tumor, recent surgery, myocardial infarction or trauma during the preceding three months, and gastrointestinal, autoimmune, inflammatory, thyroid gland, diabetes, renal, or liver disease were also excluded. Subtypes of ischemic stroke were identified based on etiology according to the TOAST classification [33]. Patients with cardioembolic (*n* = 17), large-vessel (*n* = 26), and lacunar (*n* = 25) ischemic stroke were studied. Characteristics of the ischemic stroke patients are described in Table 2. 

The study protocol conformed to the Ethical Guidelines of the World Medical Association Declaration of Helsinki and was approved by the Bioethics Committee of the University of Medical Sciences, Poznań, Poland (decision No. 661/16, approved on 12.06.2016). Written informed consent was obtained from all participants or legal guardians. The procedures followed were in accordance with institutional guidelines.

### 4.3. Serum Samples

For the CBS-deficient patients and controls, blood was drawn by venous puncture during routine health examination visits at a doctor’s office in 2012 and 2015. For the ischemic stroke patients, blood was drawn immediately after their arrival at the hospital’s emergency room in 2013 and 2014. Blood samples were allowed to clot, and serum was separated by centrifugation, collected, and stored at −80 °C.

### 4.4. Digestion with Trypsin

Firstly, 10-mg aliquots of serum protein were diluted with 15 µL of 50 mM ammonium bicarbonate, reduced with 5.5 mM dithiothreitol (5 min, 95 °C), and alkylated with 5 mM iodoacetamide (20 min in the dark, 25 °C). Serum protein was digested with Promega sequencing-grade trypsin (0.2 µg, overnight, 37 °C).

### 4.5. Label-Free Mass Spectrometry

Analyses were performed using a Dionex UltiMate 3000 RSLC nanoLC System connected to a Q Exactive Orbitrap mass spectrometer (Thermo Fisher Scientific, Roskilde, Denmark). Tryptic peptides were separated on a C18 column Acclaim PepMap RSLC nanoViper (75 µm × 25 cm; 2 µm granulation) using 4–60% acetonitrile gradient, 0.1% formic acid (300 nL/min, 30 °C, 230 min). Mass spectra were acquired on the Q Exactive in a data-dependent mode using the top 10 data-dependent MS/MS scans. The target value for the full-scan MS spectra was set to 1 × 10^6^ with a maximum injection time of 100 ms and a resolution of 70,000 at *m*/*z* 400. The MS scan range was 300–2000 *m*/*z*. The 10 most intense ions charged two or more were selected with a 2-Da isolation window and fragmented by higher-energy collisional dissociation with NCE 25. The ion target value for MS/MS was 5 × 10^4^ with a maximum injection time of 100 ms and a resolution of 17,500 at *m*/*z* 400.

Analyses of serum samples from the CBS-deficient patients (*n* = 10) and controls (*n* = 14) were carried out in duplicate and took eight days, while analyses of single samples from the ischemic stroke patients (*n* = 100) took 16.5 days. 

### 4.6. Data Analysis

Protein identification was performed using the UniProt human database (March 2017; 137,404 entries) with a precision tolerance of 10 ppm for peptide masses and 0.08 Da for fragment ion masses. Two missed trypsin cleavages were allowed. The cysteine carbamidomethylation was a fixed modification, and methionine oxidation was allowed as a variable modification. For protein identification/quantification, all raw data obtained for each dataset were imported into MaxQuant version 1.5.3.30. Protein was positively identified if at least two peptides were found by the Andromeda search engine, and a peptide score reached the significance threshold (FDR = 0.01). The analyses were based on label-free quantification (LFQ) of peptide intensities using Perseus software (version 1.4.1.3, MPIB, Martinsried, Germany). The MaxQuant data were filtered for reverse identifications (false positives), contaminants, and proteins “only identified by site”. The mean LFQ intensities ± standard deviation were calculated for each identified differentially expressed protein. The fold changes in the protein levels were assessed by comparing the mean LFQ intensities among experimental groups.

### 4.7. Statistical Analyses

The data were exported to Perseus ver. 1.5.3.2 software (part of MaxQuant package). Numeric data were log-transformed, and samples were annotated with their group affiliation. Data were filtered, and proteins with valid values in 70% of samples in at least one group were kept in the table. For multiple comparisons, one-way analysis of variance (ANOVA) with a Bonferroni correction for multiple testing was carried out. A two-sample *t*-test was used for comparisons between two groups with *p*-values < 0.05 used for truncation, and the resulting lists of differentiating proteins were normalized using the *z*-score algorithm for hierarchical clustering of data. Multivariate analyses were carried out by untargeted principal component analysis (PCA).

### 4.8. Pathway and Network Analyses

Proteins that were quantified as unique and non-redundant were used in analyses. Proteins were considered to be differentially expressed if the difference was statistically significant (*p* < 0.05). The differentiating proteins were chosen based on the criterion that the protein must be quantified by a minimum of two peptides with >99% confidence. Uncharacterized proteins were excluded from the analysis. 

Bioinformatic analysis to discover biological pathways containing proteins affected by CBS deficiency or ischemic stroke subtype, as well as networks containing those proteins, were carried out using the Ingenuity Pathway Analysis software (IPA, Ingenuity Systems, Mountain View, CA, USA). The datasets containing differentially expressed proteins were uploaded into the IPA Knowledge database, which mapped proteins/proteins products to global molecular networks to identify proteins that are known to interact with other proteins in the database.

### 4.9. ELISA Assays

To validate differentially expressed proteins, we quantified serum SAA1, SERPINC1, and GPX3 using commercial ELISA kits (Wuhan Fine Biotech Co., Ltd., Wuhan City, China). Triplicate assays were performed following the manufacturers’ protocols. A_450_ was read using an Infinite M200 Pro microplate reader (Tecan Group Ltd., Männedorf, Switzerland).

### 4.10. Anti-N-Hcy-Protein Antibody Assays

Anti-*N*-Hcy-protein antibodies were quantified via a modification of a previously described method [11]. PoliSorp 96-well plates (Thermo Fisher Scientific, Roskilde, Denmark) were coated with 0.1 mL of *N*-Hcy-albumin (10 μg/mL, 0.1 mol/L sodium carbonate buffer, pH 9.6; 37 °C, 1.5 h). Wells were washed (PBS, 0.05% Tween-20; 4 × 0.15 mL), blocked (0.1 mL of 2% BSA in PBS, 0.05% Tween-20, 1 h), and aspired to dryness. Tested human serum (0.1 mL of 50-fold diluted in PBS, 0.05% Tween-20, 0.25% BSA) was added and incubated at 4 °C for 18 h. Serum was removed, the wells were washed, and goat anti-human IgG conjugated with horseradish peroxidase (0.1 mL, 10,000-fold dilution; Millipore-Sigma, Saint Louis, MO, USA) was added. After 1 h at 37 °C, the wells were washed, and peroxidase substrate (50 μL, 1.5 mg/mL *o*-phenylenediamine, 0.03% H_2_O_2_, 0.1 mol/L citrate/phosphate buffer, pH 5.0) was added. After 1 h at 37 °C, reactions were stopped with H_2_SO_4_ (50 μL, 2 mol/L) and A_492_ was read. Each data point was obtained from triplicate measurements. Non-specific IgG binding was corrected for by subtracting A_492_ for IAA-treated *N*-Hcy-albumin controls.

### 4.11. Homocysteine Assays

Serum tHcy in CBS-deficient patients and unaffected individuals was quantified by an HPLC-based method as previously described [34]. Serum tHcy in ischemic stroke patients was quantified by mass spectrometry using a Shimadzu LCMS8060 triple quad mass spectrometer (Shimadzu Europa GmbH, Duisburg, Germany) in multiple reaction monitoring (MRM) mode by monitoring the MRM transition from *m*/*z* 136.1 to 90.1 for Hcy [35].

## Figures and Tables

**Figure 1 ijms-20-03096-f001:**
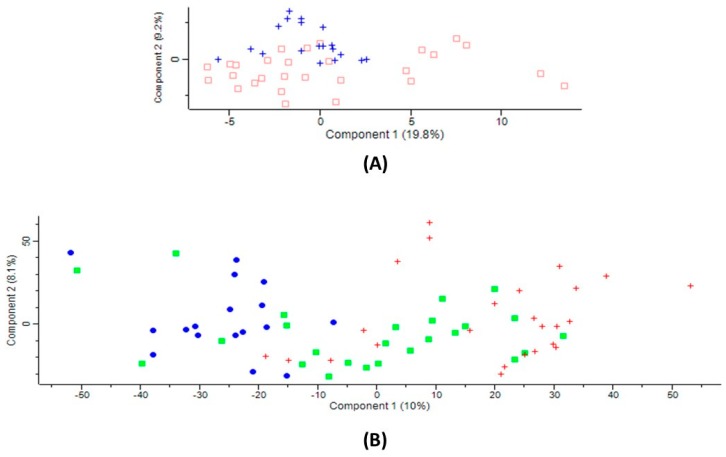
Principal component analysis (PCA) of the LFQ intensities for serum proteins: (**A**) cystathionine β-synthase (CBS)-deficient patients (blue) and unaffected controls (red); (**B**) patients with cardioembolic (blue oval), large-vessel (green square), or lacunar (red cross) ischemic stroke subtype. Calculations were carried out using Perseus.

**Figure 2 ijms-20-03096-f002:**
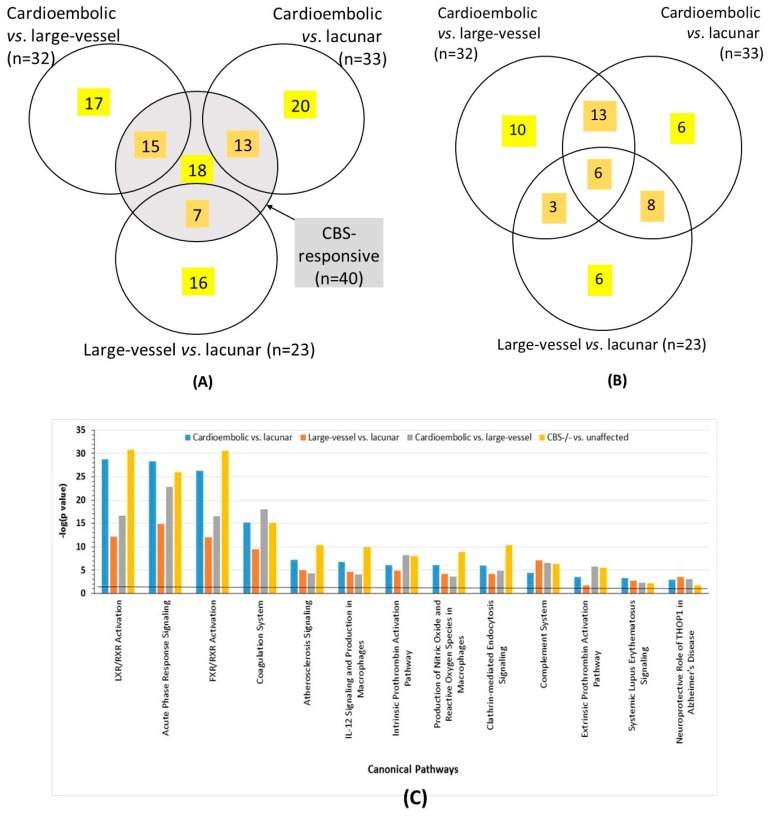
Distribution of differentially expressed serum proteins between CBS deficiency and ischemic stroke subtypes (**A**), stroke subtypes only (**B**), and their canonical pathways (**C**). (**A**,**B**) The numbers of unique (yellow) and common (gold) proteins are indicated. The middle gray circle in (**A**) represents CBS deficiency responsive proteins. The total of the numbers in the middle gray circle (*n* = 18 + 15 + 13 + 7 = 53) is greater than the number of CBS-responsive proteins (*n* = 40) because some of the proteins affected by both CBS deficiency and a stroke subtype (*n* = 22) were also affected by two or three stroke subtype comparisons (see Table 3). (**C**) Canonical pathways associated with ischemic stroke subtypes and CBS deficiency were identified by IPA. Benjamini–Hochberg, Bonferroni, and false discovery rate (FDR) corrections were applied to minimize the number of false positives.

**Figure 3 ijms-20-03096-f003:**
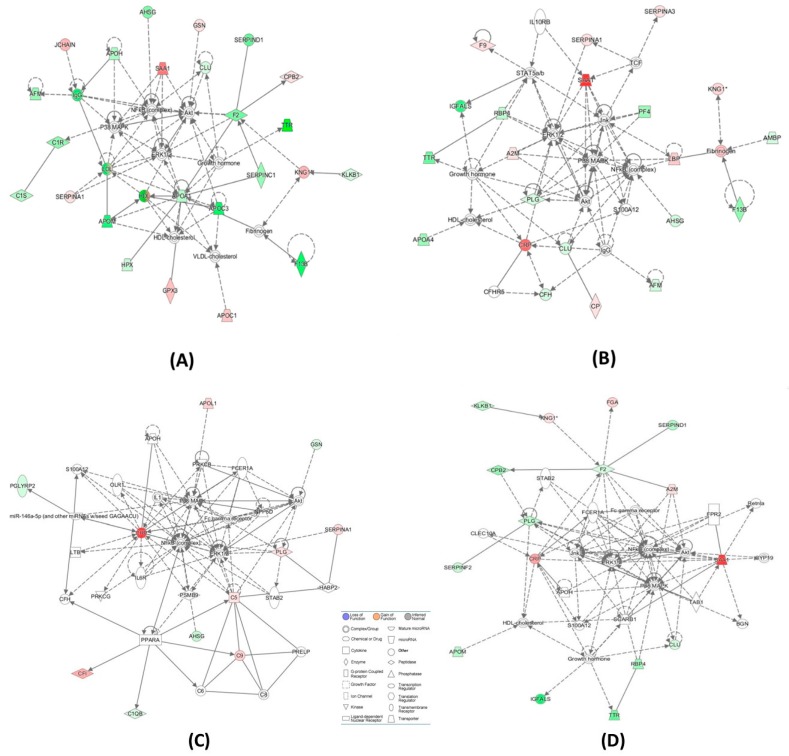
Top molecular networks associated with CBS deficiency and ischemic stroke subtypes: (**A**) CBS deficiency: humoral immune response, inflammatory response, developmental disorder; (**B**) cardioembolic vs. lacunar: cell-to-cell signaling and interaction, hematological system development and function, neurological disease; (**C**) large-vessel vs. lacunar: inflammatory response, cellular movement, immune cell trafficking; (**D**) cardioembolic vs. large-vessel: metabolic disease, hematological system development and function, lipid metabolism. Upregulated and downregulated proteins are highlighted in red and green, respectively.

**Table 1 ijms-20-03096-t001:** Characteristics of cystathionine β-synthase (CBS)-deficient patients and healthy controls.

Variable	CBS-Deficient Patients (*n* = 10)	Healthy Controls (*n* = 14)	*p*-Value
Female sex, *n* (%)	5 (50)	7 (50)	NS
Mean age, years	37.3 ± 7.3	38.0 ± 12.5	NS
History of stroke, *n* (%)	4 (40)	0 (0)	<0.05
Total cholesterol, mg/dL	180 ± 48	180 ± 37	NS
HDL cholesterol, mg/dL	58 ± 9	67 ± 22	NS
LDL cholesterol, mg/dL	193 ± 42	96 ± 35	NS
Triglyceride, mg/dL	102 ± 46	84 ± 44	NS
Methionine, µM	567 ± 16	14.9 ± 4.5	<0.001
tHcy, µM	71.2 ± 55.6	9.7 ± 5.9	<0.001
Anti-*N*-Hcy-protein antibodies, A_492_	0.33 ± 0.16	0.12 ± 0.10	0.005

**Table 2 ijms-20-03096-t002:** Characteristics of ischemic stroke patients.

Variable	All Patients (*n* = 68)	Ischemic Stroke Subtype Patients			ANOVA *p*-Value
		Large-Vessel (*n* = 28)	Cardioembolic (*n* = 16)	Lacunar (*n* = 24)	
Female sex, %	41.2	32.1	56.2	41.7	
Age, years	66.4 ± 12.7	63.8 ± 11.0	78.3 ± 15.0	62.0 ± 8.0	<0.001
Atrial fibrillation, %	34	6	91	0	
Hypertension, %	72	67	67	83	
Cholesterol, mg/dL	207.6 ± 57.4	218.0 ± 64.7	173.0 ± 64.5	211.5 ± 40.5	0.005
HDL cholesterol, mg/dL	52.0 ± 14.0	55.9 ± 14.75	51.0 ± 15.0	49.3 ± 13.0	NS
LDL cholesterol, mg/dL	131.6 ± 48.7	138.0 ± 59.0	114.5 ± 53.3	136.0 ± 33.3	NS
Triglycerides, mg/dL	116.0 ± 59.4	120.0 ± 51.0	77.8 ± 25.8	133.0 ± 72.0	0.001
Creatinine, µmol/L	94.0 ± 34.0	89.0 ± 24.7	100.6 ± 38.4	94.3 ± 34.6	NS
Glucose, mmol/L	5.85 ± 1.37	5.70 ± 1.26	5.96 ± 1.54	5.86 ± 1.36	NS
Alanine aminotransferase, U/L	24.87 ± 18.7	21.7 ± 8.6	26.8 ± 27.6	26.2 ± 13.2	NS
Aspartate aminotransferase, U/l	27.4 ± 16.4	25.9 ± 10.0	30.2 ± 23.3	26.2 ± 11.7	NS
Thyroid-stimulating hormone, mU/L	1.64 ± 1.47	1.28 ± 0.86	1.68 ± 1.08	1.94 ± 2.23	NS
Free triiodothyronine, pmol/L	5.08 ± 3.43	6.94 ± 5.4	3.76 ± 1.09	4.57 ± 0.72	0.007
Leukocytes, × 10^9^/L	8.5 ± 2.75	9.8 ± 3.4	7.3 ± 1.9	7.9 ± 2.0	0.01
tHcy, μM	3.3 ± 1.6	3.3 ± 1.2	3.5 ± 1.6	3.0 ± 2.0	NS
Anti-*N*-Hcy-protein antibodies, A_492_	0.13 ± 0.08	0.12 ± 0.08	0.14 ± 0.11	0.10 ± 0.07	NS

**Table 3 ijms-20-03096-t003:** Stroke subtype- and CBS deficiency-responsive proteins in human serum. Proteins unique for stroke subtype are highlighetd in bold.

Gene Name	Protein Name	Cardioembolic vs. Large-Vessel Stroke		Cardioembolic vs. Lacunar Stroke		Large-Vessel vs. Lacunar Stroke		ANOVA *p* Value	*CBS*^−/−^ vs. Control		Molecular Function/Biological Process
		Fold Change	*p* Value	Fold Change	*p* Value	Fold Change	*p* Value		Fold Change	T-test *p* Value	
*AFM*	Afamin	0.84	0.011	0.85	0.005			0.0148	0.83	0.007	Vitamin transport
*ORM1*	α-1-acid glycoprotein1			1.35	0.001	1.20	0.010	0.0018			Acute inflammatory response/acute-phase response
*ORM2*	α-1-acid glycoprotein2								0.81	0.042	Acute-phase response
*SERPINA3*	α-1-antichymotrypsin	1.21	0.046	1.42	<1 × 10^−4^	1.18	0.021	0.0002			Acute inflammatory response/acute-phase response
*SERPINA1*	α-1-antitrypsin			1.17	0.007	1.18	0.011	0.0106	1.14	0.045	Acute phase response; Blood coagulation
*SERPINF2*	**α-2-antiplasmin**	**0.85**	**0.027**					0.0305	0.80	1.2 × 10^−5^	Complement/coagulation cascades
*AHSG*	α-2-HS-glycoprotein			0.85	0.001	0.81	0.0001	0.0001	0.80	5.3 × 10^−5^	Acute phase response
*SERPINC1*	Antithrombin-III								0.84	4.5 × 10^−5^	Complement/coagulation cascades
*A2M*	α-2-macroglobulin	1.29	0.024	1.29	0.006						Blood coagulation
*APOA1*	Apolipoprotein A-I								0.87	0.038	Fat digestion/absorption
*APOA4*	**Apolipoprotein A-IV**			**0.71**	**0.006**						Cholesterol transport
*APOC1*	Apolipoprotein C-I	0.67	0.004	0.52	<1 × 10^−4^	0.77	0.035	4.5E-5	1.40	0.032	Cholesterol efflux/lipid and lipoprotein metabolic process
*APOC3*	Apolipoprotein C-III								0.68	0.014	Cholesterol transport
*APOL1*	**Apolipoprotein L1**					1.23	0.042				Cholesterol metabolism
*APOM*	**Apolipoprotein M**	0.81	0.012					0.0375	0.70	0.001	Cholesterol transport, antioxidant activity
*APOH*	Beta-2-glycoprotein 1T								0.86	0.015	Blood coagulation
*C4BPA*	**C4b-binding protein α-chain**	0.44	0.032								Complement activation/immune response
*CPB2*	**Carboxypeptidase B2**	**0.74**	**0.018**					0.0431	1.12	0.044	Complement/coagulation cascades; Protein digestion
*CP*	Ceruloplasmin			1.21	0.001	1.12	0.007	0.0020			Cellular iron ion homeostasis
*CLU*	Clusterin	0.88	0.014	0.88	0.004			0.0072	0.93	0.019	Negative regulation of amyloid-beta formation
*F9;factor IX*	Coagulation factor IX			1.45	0.004	1.29	0.018	0.0010			Blood coagulation
*F13B*	Coagulation factor XIII B-chain	0.74	0.010	0.71	0.005			0.0010	0.69	0.0005	Blood coagulation
*C1QA*	**Complement C1q subunit A**	**1.66**	**0.042**								Complement and coagulation cascades
*C1QB*	Complement C1q subunit B	1.12	0.038			0.90	0.047	0.0433			Complement/coagulation cascades
*C5*	**Complement C5**					**1.14**	**0.003**	0.0136			Complement/coagulation cascades
C1R	Complement C1r subcomponent								0.82	5 × 10^−5^	Complement/coagulation cascades
C1S	Complement C1s subcomponent								0.86	0.004	Complement/coagulation cascades
*C9*	Complement component C9			1.36	1 × 10^−4^	1.25	0.002	0.0002	1.36	0.004	Complement/coagulation cascades
*CFI*	Complement factor I								0.85	0.002	Complement/coagulation cascades
*CRP*	C-reactive protein	2.44	0.046	5.89	1 × 10^−5^	2.41	0.011	2.5 × 10^−7^			Acute inflammatory response/acute-phase response
*HEL-213*	Epididymis luminal protein 213								1.55	0.003	
*FGA*	Fibrinogen α-chain;	1.53	0.006	2.01	1 × 10^−6^	1.32	0.031	1.6 × 10^−5^			Blood coagulation
*FBLN1*	Fibulin-1	**1.99**	**0.0005**					0.0326	2.13	0.001	Blood coagulation/fibrin clot formation
*FCN3*	Ficolin-3			**0.72**	**0.007**			0.0211	0.76	0.003	Complement activation
*GSN*	Gelsolin					**0.87**	**0.004**	0.0160	1.11	0.015	Actin filament capping/Amyloid fibril formation
*GPX3*	**Glutathione peroxidase 3**					**1.26**	**0.018**		1.41	0.005	Cellular response to oxidative stress
*HPR*	Haptoglobin-related	0.61	0.002	0.64	0.001			0.0018			Receptor-mediated endocytosis
*HPX*	Hemopexin								0.90	0.004	Cellular iron ion homeostasis
*SERPIND1*	Heparin cofactor 2	0.80	0.027	0.71	4 × 10^−4^			0.0023	0.77	0.005	Complement/coagulation cascades
*H2AFJ*	**Histone H2A**					**0.49**	**0.027**				Chromatin silencing
*HIST1H4A*	Histone H4			0.35	0.001	0.43	0.003	1.2 × 10^−5^			Telomere organization
IGHV3-7	Ig heavy chain V-III region GAL								1.64	0.019	Immune response
*IGKV1D-12*	**IgK chain V-I region Wes**	**0.77**	**0.032**								Immune response
*IGH@*	IGH@ protein								0.71	0.006	Immune response
*IGK@*	**IGK@ protein**					**0.88**	**0.019**		1.31	0.002	Immune response
IGHD	Immunoglobulin heavy constant delta								3.39	0.009	Immune response
IGHV3-72	Ig heavy variable 3-72								1.48	0.025	Immune response
IGJ; JCHAIN	Ig J-chain								1.51	0.006	Immune response
IGKV2D-24	IgK variable								0.70	0.024	Immune response
*IGFALS*	Insulin-like growth fac-tor-binding complex acid-labile subunit	0.59	0.003	0.58	0.002			0.0005			Cell adhesion
*ITIH2*	Inter-α-trypsin inhibi-tor heavy chain H2								0.92	0.022	Amine metabolic process
*ITIH3*	Inter-α-trypsin inhibi-tor heavy chain H3	1.46	0.001	1.72	1 × 10^−5^			1.1 × 10^−5^			Amine metabolic process
*ITIH4*	**Inter-α-trypsin inhi-bitor heavy chain H4**			**1.12**	**0.001**						Amine metabolic process
N/A	cDNA FLJ53075, high-ly similar to KNG1								1.55	0.001	
*SERPINA4*	Kallistatin	0.70	0.012	0.59	<1 × 10^−4^	0.84	0.018	4.8 × 10^−5^			Platelet degranulation
*KNG1*	Kininogen-1	0.92	0.029	0.92	0.009			0.0002	0.86	0.000	Blood coagulation/inflammatory response
*LBP*	**Lipopolysaccharide-binding protein**			**1.90**	**0.002**						Acute phase response
*LUM*	Lumican	1.27	0.017			0.78	0.002	0.0037			Collagen binding
*PGLYRP2*	*N*-acetylmuramoyl-L-alanine amidase			0.87	0.001	0.91	0.023	0.0055			Immune resonse/inflammatory response
*KLKB1*	**Plasma kallikrein**	**0.81**	**0.016**						0.91	0.043	Complement/coagulation cascades
*PLG*	Plasminogen	0.82	0.0002	0.89	0.004	1.09	0.039	1 × 10^−4^			Blood coagulation, fibrynolysis
*PF4*	**Platelet factor 4**			**0.72**	**0.001**			0.0264			Platelet degranulation, inflammatory response
*PZP*	Pregnancy zone protein	3.87	0.0003			0.38	0.004	0.0001			Female pregnancy
*AMBP*	**Protein AMBP;** **α-1-microglobulin; Trypstatin**			**0.90**	**0.005**			0.0497			Cell adhesion; Heme metabolic process
*F2*	**Prothrombin**	**0.90**	**0.008**					0.0142	0.79	8 × 10^−6^	Complement/coagulation cascades; Neuroactive ligand-receptor interaction; Regulation of actin cytoskeleton
*RBP4*	Retinol-binding protein 4	0.75	0.010	0.79	0.012			0.0093			Cardiac muscle tissue development
*SAA1*	Serum amyloid A-1	6.31	0.004	11.13	<1 × 10^−5^			0.0052	1.97	0.020	Acute-phase response
*APCS*	**Serum amyloid P**	**0.83**	**0.010**					0.0237			Immune response
*HEL111;TTR*	Transthyretin	0.68	0.004	0.64	0.004			0.0060	0.57	5.4 × 10^−9^	Retinol metabolic process, thyroid hormone transport
*HEL-S-51;GC*	Vit. D-binding protein	0.91	0.020	0.87	<1 × 10^−3^			0.0007	0.85	0.0001	Vitamin D metabolic process

**Table 4 ijms-20-03096-t004:** List of stroke subtype- and CBS deficiency-specific proteins.

Cardioembolic vs. Large-Vessel Stroke	Cardioembolic vs. Lacunar Stroke	Large-Vessel vs. Lacunar Stroke	CBS^−/−^ vs. Control
APCS	AMBP	APOL1	APOA1
APOM	APOA4	C5	APOC3
C1QA	FCN3	GSN	APOH
C4BPA	ITIH4	GPX3	C1R
CPB2	LBP	H2AFJ	C1S
FBLN1	PF4	IGK@	CFI
IGKV1D-12			HEL0213
KLKB1			HPX
SERPINF2			IGHV3-7
			IGHD
			IGHV3-7
			IGH@
			IGJ; JCHAIN
			IGKV2D-24
			ITIH2
			ORM2
			SERPINC1
			cDNA FLJ53075, highly similar to KNG1

**Table 5 ijms-20-03096-t005:** Top molecular networks of stroke subtype-responsive and CBS deficiency-responsive proteins in the human serum. Upregulated (↑) and downregulated (↓) proteins are highlighed in red and green, respectively. Graphical illustrations of interactions between proteins in these networks are shown in Figure 3A–D.

Analysis	Molecules in Network	Score	Focus Molecules	Top Diseases and Functions
**Cardioembolic vs. large-vessel stroke** **(Figure 3A)**	↑A2M, Akt, APOH, ↓APOM, BGN, CLEC10A, ↓CLU, ↓CPB2,↑CRP, CYP19, ERK1/2, ↓F2, Fc gamma receptor, FCER1A, ↑FGA, FPR2, Growth hormone, HDL-cholesterol, ↓IGFALS, Jnk, KLKB1,↑KNG1, NFkB (complex), P38 MAPK, ↓PLG, ↓RBP4, Retnla, S100A12, ↑SAA1, SCARB1, ↓SERPIND1, ↓SERPINF2, STAB2, TAB1, ↓TTR	36	16	Hematological System Development and Function, Cell-To-Cell Signaling and Interaction, Organismal Functions
**Cardioembolic vs. large-vessel stroke**	↓AFM,ALT,↓APCS, ↓APOC1,APOE,C6, ↑C7,C8,C9,C1q,↑C1QA, ↑C1QB, ↓C4BPA, Ccl2, ↑CRP, CRYAB, CTSB, CXCL2, ↓F13B,↑FBLN1, ↑FGA,↓GC, HNF1A, HNF4A, HRG, IL1R1, INSR, ↑ITIH3, LDL-cholesterol, ↑LUM, PSEN2, SCARB1, ↑SERPINA3, STK40, VLDL-cholesterol	33	15	Metabolic Disease, Developmental Disorder, Hereditary Disorder
**Cardioembolic vs. lacunar stroke** **(Figure 3B)**	↑A2M, ↓AFM, ↓AHSG, Akt, ↓AMBP, ↓APOA4, ↓CFH,CFHR5, ↓CLU, ↑CP, ↑CRP, ERK1/2, ↑F9, ↓F13B, Fibrinogen, Growth hormone, HDL-cholesterol, ↓IGFALS, IgG, IL10RB, Jnk, ↑KNG1, ↑LBP, NFkB (complex), P38 MAPK, ↓PF4, ↓PLG, ↓RBP4,S100A12, ↑SAA1, ↑SERPINA1, ↑SERPINA3,STAT5a/b, TCF, ↓TTR	50	21	Cell-To-Cell Signaling and Interaction, Hematological System Development and Function, Neurological Disease
**Cardioembolic vs. lacunar stroke**	↓AHSG, ↓APOC1, ARG1, C5, C6,↑C7, C8,↑C9, CCND1, CPB2, ↑CRP, CTSB, EHF, ↓FCN3, MASP1, MASP2, Fc-γ receptor, FGA, FGB, FGG, ↓GC, ↓HIST1H4H, HNF1A, HNF4A, IL6, ↑ITIH3, ↑ITIH4, LDL-cholesterol, LGALS3, N-cor, NR1H2, NR5A2, ↑ORM1, PGLYRP2, PRELP, ↑SAA1, ↓SERPIND1, TTR	32	15	Inflammatory Response, Metabolic Disease, Cell-To-Cell Signaling and Interaction
**Large-vessel vs. lacunar stroke** **(Figure 3C)**	↓AHSG, Akt, APOH, ↑APOL1, ↑C5, C6, C8, ↑C9, ↓C1QB, CFH, ↑CFI, ↑CRP, ERK1/2, Fc-γ receptor, FCER1A, ↓GSN, HABP2, IL1, IL6R, INPP5D, LTB, miR-146a-5p (and other miRNAs w/seed GAGAACU), NFkB (complex), OLR1, P38 MAPK, ↓PGLYRP2, ↑PLG, PPARA, PRELP, PRKCB, PRKCG, PSMB9, S100A12, ↑SERPINA1, STAB2	24	11	Inflammatory Response, Cellular Movement, Immune Cell Trafficking
**Large-vessel vs. lacunar stroke**	AGER, ↓APOC1, CCND1, CDK4, ↑CP, CRYAB, EHF, F2, ↑F9, Fc-γ receptor, FCER1A, Ferritin, ↑FGA, FGB, FGG, ↑GPX3, ↓H2AFJ, ↓HIST1H4H, INPP5D, ↑KNG1, Ldh (complex), LRPAP1, ↓LUM, MAPK1,N-cor, NR5A2, Nuclear factor 1, ↑ORM1,PLCG2, PSMB9, PSME2, ↑SERPINA3, TNF, TNFSF10, TP53	24	11	Cell-To-Cell Signaling and Interaction, Hematological System Development and Function, Inflammatory Response
**CBS^−/−^ vs. control** **(Figure 3D)**	↓AFM, ↓AHSG, Akt, ↓APOA1,↑APOC1,↓APOC3, ↓APOH, ↓APOM, ↓C1R, ↓C1S, ↓CLU,↑CPB2,ERK1/2,↓F2, ↓F13B, Fibrinogen, ↑GPX3, Growth hormone, ↑GSN, HDL, HDL-cholesterol, ↓HPX, IgG, ↑JCHAIN, ↓KLKB1, ↑KNG1, LDL, NFkB (complex), P38 MAPK, ↑SAA1, ↑SERPINA1, ↓SERPINC1, ↓SERPIND1, ↓TTR, VLDL-cholesterol	60	24	Metabolic Disease, Hematological System Development and Function, Lipid Metabolism
**CBS^−/−^ vs. control**	ADAMTS1, ↓AHSG,ALT, ↓APOC3, BGN, BIRC5, C6, C7,↑C9, ↓C1S,CCND1, ↓CFI, ↑CPB2, ↓CPN1, F13A1, ↑FBLN1, ↓FCN3, FGA, FPR2, ↓GC, GCNT3, GPR119, HNF1A, ↓IGHG1, IL1B, ↓ITIH2, MASP1, MASP2, ↓ORM2, PPARA, ↑SAA1, ↓SERPIND1, ↓SERPINF2, TG, TNF	35	16	Humoral Immune Response, Inflammato-ry Response, Develop-mental Disorder

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
