# Peer review of "Serum Proteome Alterations in Human Cystathionine β-Synthase Deficiency and Ischemic Stroke Subtypes"

_ijms, 2019, doi:10.3390/ijms20123096_

Round 1
Reviewer 1 Report
This manuscript uses the label-freemass spectromtery to quantify changes in serum proteomes in cystathionine beta-sytnhase (CBS)-deficient patients and unaffected controls as well as in patients with cardioembolic, large-vessel or lacunar stroke. Significant overlaps were observed between proteins affected by CBS deficiency and ischemic stroke subtypes, similar to protein overlaps between ischemic stroke subtypes. Form these findings the Authors infer common mechanisms involved in the pathologies of CBS-deficiency and ischemic stroke subtypes..The study is clear, and methods are adequate. However the interest of results appears to be limited, and the conclusions regarding common mechanisms appear to be arbitrarily obtained from similarities in serum proteomes of different subject groups. On the basis of the present considerations. while methodological concerns are essentially lacking in this descriptive study, the priority and the interest appear to be reduced. Of course the final decision, on the basis of the policy of the journal, pertains to the Editor.
Author Response
We thank Reviewer 1 for the comments.
We have substantially modified the Discussion to address the Reviewer's comment.
Reviewer 2 Report
I think that what appears to be the overall aim of this paper, i.e., to identify proteomic similarities and differences between ischemic stroke subtypes and another disease process (CBS-deficiency) is relevant and of significant potential interest. The methodology of mass spect and bioinformatics analysis is promising. However, the manuscript presents this information in a somewhat confusing and unclear way.
The abstract presents very few of the actual results that would be beneficial for the reader. All of the bioinformatics-related results, such as acute phase response signaling, are contained in 3 of the final 4 lines of the abstract. The abstract should more explicitly state that this study only includes ischemic stroke. The abstract should also include information about the study's sample sizes.
The introduction is poorly written. Ischemic and hemorrhagic stroke should not both be included if the manuscript only presents results from ischemic. The background information, e.g, "stroke is a leading cause of morbidity and mortality in the world" is too general and does not include any citation. Also, the unclear and potentially inaccurate statement "each stroke subtype is treated by a different therapy." To what type of treatment are the authors referring? Acute stroke treatment? In fact, both large vessel and cardioembolic strokes could be treated with IV tPA and thrombectomy acutely, so it is not accurate to state that they are treated by a different therapy.
I also do not think that the authors make a clear case in the introduction for why it is beneficial to study the proteome in ischemic stroke patients vs. CBS-deficient patients. Why not just study ischemic stroke subtypes by themselves? Why not study patients with other hypercoagulable states, i.e., not CBS-deficiency? What background/references would support WHY the reader should want to know about CBS-deficiency vs ischemic stroke?
Overall, with all due respect, the introduction does not give the impression that that authors have a strong grasp of stroke as a disease process and the associated current treatments. More concise and detailed information, including appropriate references, is needed.
In section 4.2, it is not clear why the section begins with "ischemic stroke patients" as the overall "n =103," but then describes the exclusion of hemorrhagic and unknown stroke. Then ultimately, the final sample size for this study is 17+26+25= 68? I would recommend only presenting the ischemic stroke patients that were included, their demographics, and the inclusion/exclusion criteria for ischemic stroke.
Typically, the results for a study such as this will be complicated and include long lists of proteins, tables with a large numbers of proteins, complex bioinformatics-generated networks, etc. For such a study, it is very important that the discussion section of the manuscript clarifies the main points that the authors want the reader to take away. As currently written, the discussion does not do this. Instead, it presents a large number of paragraphs, many seeming to present a single part of the results of the manuscript, but not a main concept. Each main section/paragraph of the discussion should begin with a major concept that the reader should take away from the results the current manuscript, and then that concept is expanded in the context of other references.
For example, paragraph 1 of the discussion begins by unnecessarily presenting a long list of results of specific proteins. What is the main point of paragraph 1 of the discussion? That CBS-deficiency and ischemic stroke share similar molecular mechanisms? If so, then state that at the beginning, something like, "Our results supported that CBS-deficiency and ischemic stroke share similar molecular mechanisms" and then expand on that in the context of the current paper and supporting literature. The reader should not have to search for the main points of the discussion; as it is currently written, this searching is necessary.
It does not appear that a conclusion section is required for this paper, but some form of summary paragraph for the entire study and future directions is very much needed.
Author Response
IJMS Reviewer 2 comments:
I think that what appears to be the overall aim of this paper, i.e., to identify proteomic similarities and differences between ischemic stroke subtypes and another disease process (CBS-deficiency) is relevant and of significant potential interest. The methodology of mass spect and bioinformatics analysis is promising. However, the manuscript presents this information in a somewhat confusing and unclear way.
The abstract presents very few of the actual results that would be beneficial for the reader. All of the bioinformatics-related results, such as acute phase response signaling, are contained in 3 of the final 4 lines of the abstract. The abstract should more explicitly state that this study only includes ischemic stroke. The abstract should also include information about the study's sample sizes.
Response: The Abstract states that the study includes ischemic stroke and CBS deficiency patients. The information on the number of participants in each study group has now been included.
The introduction is poorly written. Ischemic and hemorrhagic stroke should not both be included if the manuscript only presents results from ischemic. The background information, e.g, "stroke is a leading cause of morbidity and mortality in the world" is too general and does not include any citation. Also, the unclear and potentially inaccurate statement "each stroke subtype is treated by a different therapy." To what type of treatment are the authors referring? Acute stroke treatment? In fact, both large vessel and cardioembolic strokes could be treated with IV tPA and thrombectomy acutely, so it is not accurate to state that they are treated by a different therapy.
Response: A reference for the statement "Stroke is a leading cause of morbidity and mortality in the world” is now included. The statement "each stroke type/subtype is treated by a different therapy" has been deleted and the 1st paragraph of the Introduction has been re-written.
I also do not think that the authors make a clear case in the introduction for why it is beneficial to study the proteome in ischemic stroke patients vs. CBS-deficient patients. Why not just study ischemic stroke subtypes by themselves? Why not study patients with other hypercoagulable states, i.e., not CBS-deficiency? What background/references would support WHY the reader should want to know about CBS-deficiency vs ischemic stroke?
Response: We believe the high prevalence of ischemic strokes in CBS-deficient patients justifies studying them vs. ischemic stroke patients in the general population in order to elucidate possible mechanisms involved. This is highlighted in the 2nd paragraph of the Introduction, lines 50-58.
Overall, with all due respect, the introduction does not give the impression that that authors have a strong grasp of stroke as a disease process and the associated current treatments. More concise and detailed information, including appropriate references, is needed.
Response: The Introduction has been modified as suggested.
In section 4.2, it is not clear why the section begins with "ischemic stroke patients" as the overall "n =103," but then describes the exclusion of hemorrhagic and unknown stroke. Then ultimately, the final sample size for this study is 17+26+25= 68? I would recommend only presenting the ischemic stroke patients that were included, their demographics, and the inclusion/exclusion criteria for ischemic stroke.
Response: The ischemic stroke patients that were included in the study, their demographics, and the inclusion/exclusion criteria for ischemic stroke are now presented in section 4.2., lines 366-379.
Typically, the results for a study such as this will be complicated and include long lists of proteins, tables with a large numbers of proteins, complex bioinformatics-generated networks, etc. For such a study, it is very important that the discussion section of the manuscript clarifies the main points that the authors want the reader to take away. As currently written, the discussion does not do this. Instead, it presents a large number of paragraphs, many seeming to present a single part of the results of the manuscript, but not a main concept. Each main section/paragraph of the discussion should begin with a major concept that the reader should take away from the results the current manuscript, and then that concept is expanded in the context of other references.
For example, paragraph 1 of the discussion begins by unnecessarily presenting a long list of results of specific proteins. What is the main point of paragraph 1 of the discussion? That CBS-deficiency and ischemic stroke share similar molecular mechanisms? If so, then state that at the beginning, something like, "Our results supported that CBS-deficiency and ischemic stroke share similar molecular mechanisms" and then expand on that in the context of the current paper and supporting literature. The reader should not have to search for the main points of the discussion; as it is currently written, this searching is necessary.
Response: The discussion has been modified throughout as suggested.
It does not appear that a conclusion section is required for this paper, but some form of summary paragraph for the entire study and future directions is very much needed.
Response: A summary paragraph has been included as suggested, lines 351-355.
Round 2
Reviewer 2 Report
The authors have responded adequately to the comments that I submitted initially. I do not have any suggestions for additional edits at this time.